# Radiology Community Attitude in Saudi Arabia about the Applications of Artificial Intelligence in Radiology

**DOI:** 10.3390/healthcare9070834

**Published:** 2021-07-01

**Authors:** Magbool Alelyani, Sultan Alamri, Mohammed S. Alqahtani, Alamin Musa, Hajar Almater, Nada Alqahtani, Fay Alshahrani, Salem Alelyani

**Affiliations:** 1Department Radiological Sciences, King Khalid University, Abha 61421, Saudi Arabia; mosalqhtani@kku.edu.sa (M.S.A.); aomer@kku.edu.sa (A.M.); 437804506@kku.edu.sa (H.A.); 436803611@kku.edu.sa (N.A.); 437804448@kku.edu.sa (F.A.); 2Department Radiological Sciences, Taif University, Taif 21944, Saudi Arabia; s.alamri@tu.edu.sa; 3Center for Artificial Intelligence (CAI), King Khalid University, Abha 61421, Saudi Arabia; s.alelyani@kku.edu.sa; 4College of Computer Science, King Khalid University, Abha 61421, Saudi Arabia

**Keywords:** AI, radiology, awareness, radiographers, radiologists

## Abstract

Artificial intelligence (AI) is a broad, umbrella term that encompasses the theory and development of computer systems able to perform tasks normally requiring human intelligence. The aim of this study is to assess the radiology community’s attitude in Saudi Arabia toward the applications of AI. Methods: Data for this study were collected using electronic questionnaires in 2019 and 2020. The study included a total of 714 participants. Data analysis was performed using SPSS Statistics (version 25). Results: The majority of the participants (61.2%) had read or heard about the role of AI in radiology. We also found that radiologists had statistically different responses and tended to read more about AI compared to all other specialists. In addition, 82% of the participants thought that AI must be included in the curriculum of medical and allied health colleges, and 86% of the participants agreed that AI would be essential in the future. Even though human–machine interaction was considered to be one of the most important skills in the future, 89% of the participants thought that it would never replace radiologists. Conclusion: Because AI plays a vital role in radiology, it is important to ensure that radiologists and radiographers have at least a minimum understanding of the technology. Our finding shows an acceptable level of knowledge regarding AI technology and that AI applications should be included in the curriculum of the medical and health sciences colleges.

## 1. Introduction

Artificial intelligence (AI) is a broad, umbrella term that encompasses the theory and development of computer systems able to perform tasks normally requiring human intelligence [1]. In several fields including healthcare, AI is moving quickly from an experimental phase to an implementation phase [2]. The word “AI” includes the sciences and innovations that use computers to simulate, expand, or even enhance human intelligence. AI is directly related to the information technological revolution, cognitive science, analytics, and algorithms [3]. In the best-case scenario, AI algorithms will provide an additional tool for radiologists, close to a “second pair of eyes”, giving an additional point of view on cases and improving competency and diagnostic reliability. This is the equivalent of a radiologist asking a colleague whom he or she trusts for a second point of view about a case [4]. In addition, the implementation of AI in medical imaging needs radiological technologists to further adapt with the integration between AI and medical imaging. Therefore, treatment practice and imaging should be enhanced with new technology, as high-quality practice and research will provide benefits to patients [5].

In regard to the imaging reports, there is specialized terminology to describe radiographic appearances precisely, which radiologists share with others within their profession, as well as with referring physicians and patients. As AI technology expands and ultimately becomes a part of the clinical workflow, radiologists need to familiarize themselves with its basic principles and terminology [6]. Medicine and, more specifically, radiology are witnessing continuous changes associated with AI and machine learning [7]. Recently, AI technologies for radiology applications have gained popularity among healthcare providers [8], due to the fact that radiology is one of the most prolific generators of a huge amount of digital data [9], leading to increased work pressures for specialists and radiologists. Therefore, there is a growing need to develop technologies that carry some of the workload [10].

Extensive research has shown that AI technology assists with image recognition and acquisition, improves the support tools for radiology decisions, helps monitor possible diagnostic errors, and facilitates intelligent scheduling solutions [7,11]. The first step toward enhancing the application of AI in radiology requires a deep knowledge of current capabilities and future concerns [12]. However, the radiology community needs to be trained about how to critically analyze the possibilities, risks, and threats associated with the implementation of new AI instruments [2]. In all of the studies reviewed here, no progress could be made in applications of AI related to radiology and medical imaging in Saudi Arabia without surveying the readiness of the workforce in radiology. The role of radiologists in the AI era is to become expert consumers of AI algorithms. There are opportunities to capitalize on the latest emphasis and excitement surrounding modern AI technologies and chances to use data to educate radiologists about their possibilities. The revolution of AI growth is gaining momentum, and the variety of AI applications being presented to radiologists is growing, providing challenges to radiologists about the best tools to select [4].

The most beneficial AI tools for radiologists are the applications that will best fulfil their clinical needs and help them to answer the questions asked by their referring physicians [3]. The radiologists’ practice may be enhanced using AI tools, which may affect the their clinical experience [4]. Focusing on the radiologists’ knowledge and training on the use of the AI tools is very important, as this assists their practice, which, in turn, benefits patients [4]. Radiologists should collaborate with the computer science and engineering departments of their respective universities and contribute to the research and development of AI in healthcare systems to guarantee that there is full clinical value to the problems under examination [2].

Moreover, there are many areas in medical imaging that show the direct impact of AI on the radiographers’ role, such as pre-examination assessment, examination planning, imaging acquisition, and image processing. For example, radiographers usually contact patients directly before imaging to explain the procedures and take care of them after imaging. This is unlikely to be changed by AI technologies. However, there is the potential for AI to contribute to and help in the automated examinations of referrals, checking the clinical indications and confirming the patient’s identification via an interface with the Electronic Health Record. AI technologies can also assist in imaging modality techniques and processes [13].

Currently, there are no data to measure the amount of awareness about AI in Saudi Arabia. To address this issue, we performed an electronic survey to assess the radiology community’s attitude toward AI applications in Saudi Arabia. This would improve our knowledge on the future direction of choosing radiology as a lifetime career.

## 2. Materials and Methods

Data for this study were collected using an electronic questionnaire using the Google Forms application, and the link was distributed across Saudi Arabia through emails, WhatsApp groups, and Twitter during the period from 2019 and 2020. A total of 714 participants were included in the study. The target group in this study was radiologists, radiology technologists, technicians, and radiological sciences students from different regions around Saudi Arabia. The questionnaire consisted of two parts. The first part was related to demographic data that contained gender, age, highest qualification attained, and subspecialty. The second part was structured into eight sections with a total of 25 statements. Each section addressed different aspects of AI technology (see Appendix). The first section contained questions on the participants’ awareness of AI, and section two covered AI applications in clinical practice. Sections three to five dealt with AI results, AI responsibilities, and AI validation, respectively. Section six focused on the role of the patient, whereas section seven focused on the benefits of AI in medicine. The last section was related to the future of AI.

A three-point Likert scale was used, and the participants were asked to answer each question using one of the following options: (Agree, Neutral, Disagree). To simplify the analysis, the neutral responses were considered negative and ultimately were regarded as a disagreement with the question. Data management and analysis were performed using SPSS Statistics (version 25). This study did not require ethical approval because there was no risk to the participants. A one-way ANOVA and chi square were used to compare the differences in the awareness of the participants. When there was no homogeneity of variance, and in order to minimize Type 1 error, the more conservative Welch’s *t*-test was applied.

## 3. Results

### 3.1. Descriptive Data Analysis

A total of 714 individuals participated in the study. Of them, 45.4% (N = 324) were female and 54.6% (N = 390) were male. The age distribution of participants is shown in Figure 1. The inclusion of participants with varying age allowed for the assessment of differing experiences of individuals, as experience might change with age. The majority of participants (n = 245; 34%) belonged to the age range of 20 to 25 years. This was followed by participants belonging to the age range of 26 to 30 years (n = 172; 24%), 31 to 35 years (n = 123; 17%), and more than 40 years (n = 83; 12%).

In the present study, the qualifications of the participants were distributed by a breakdown of 44.7% (N = 319) with a bachelor’s degree, 6.9% (N = 49) with a diploma, 15% (N = 107) with a masters, 14.6% (N = 104) with a Ph.D., and 135 (18.9%) of them were undergraduate students. Through the inclusion of participants with different educational levels, the relevant attitudes could be understood. The participants’ profession and locations are graphically represented in Figure 2 and Figure 3, respectively.

### 3.2. AI Awareness

The first set of questions aimed to evaluate the participants’ awareness of AI in general, and its uses in radiology in particular (Table 1). It is clear from Table 1 that the majority of the participants (61.2%) have read or heard about the role of AI in radiology. Interestingly, the odds of male participants reading or hearing about the role of AI in radiology is significantly higher than that for female participants (odd ratio (OR): 2.4, 95% confidence interval (CI): 1.76 to 3.3, *p* < 0.001). Moreover, participants who are more than 30 years of age are more likely to have read or heard about AI in radiology (OR: 0.47, CI: 0.34 to 0.64, *p* < 0.001). In addition, there was a significant difference between the group means according to their education level as determined by one-way ANOVA (Welch’s F (2, 426.7) = 46.4, *p* < 0.005). From this, we can infer that participants with a high level of education are more likely to have read or heard about AI. We also observed that radiologists have statistically different responses and tend to read more about AI compared to all other professions, (Welch’s F (4, 709) = 24, *p* < 0.005).

In addition, less than half the participants (44%) were keen to attend conferences or courses about AI in radiology. A closer inspection showed that the females as well as younger participants were less likely to attend conferences compared with the males and the older group, (OR: 1.6, CI: 1.2 to 2.2, *p* < 0.005 and OR: 0.4, CI: 0.33 to 0.6, *p* < 0.001, respectively). Surprisingly, participants who specialized in CT/MRI were more eager to attend conferences than radiologists were, (Welch’s F (4, 341) = 9.4, *p* < 0.001).

Moreover, male and older groups were almost twice as more likely to be involved in research involving AI compared with the female and younger groups, (OR: 2, CI: 1.4 to 2.0, *p* < 0.001, OR: 0.55, CI: 0.39 to 0.8, *p* < 0.005, respectively). In addition, we found no significant association between the profession and location in regard to the knowledge about the contribution of AI in the preparation of radiographic reports, *p >* 0.05.

### 3.3. AI Practices

The second set of questions assessed the participants’ knowledge of the practices of AI in radiology (Table 2). The results showed that the vast majority of people (71.3%) agreed with the statement that AI contributed to the capture of high-quality images. Our statistical analysis showed that both the gender and age had no effect on the participants’ decision on this question, *p >* 0.5. Furthermore, radiologists significantly differed from college students in that they were more likely to disagree with the use of AI in the selection of scanning protocols for CT and MRI, *p* < 0.05. Interestingly, although the majority of participants believed that AI could help in obtaining high-quality images and assist with the archiving system, only about half felt that their jobs were threatened by this technology. What was interesting about the data in this table was that the participants’ education levels and locations did not appear to have an effect on their responses.

### 3.4. AI Outcomes

The third set of questions assessed the participants’ knowledge of the outcomes of AI in radiology (Table 3). Closer inspection of Table 3 showed that 60% of the participants felt the results produced by AI are not reliable. The belief was more common in B.Sc. holders and higher education levels compared with that in diploma holders and college students, (Welch’s F (2, 717.9) = 7.67, *p* < 0.01). Even though 80% of the participants thought that the results produced by AI should be verified by radiologists, this was found to be significant between the participating radiologists and other professionals, (Welch’s F (4, 434.4) = 8.02, *p* < 0.001). When the participants were asked about whether AI causes stress and anxiety to the patients, 60% of them agreed. Further analysis showed that the younger group were more likely to agree with this statement, (OR: 1.9, CI: 1 to 1.9, *p* < 0.05). Finally, the majority of participants thought that radiologists should not have to bear the responsibility of the results obtained by utilizing AI but that it should be shared between AI companies and organizations, 75% and 83%, respectively.

### 3.5. AI Responsibilities

The fourth set of questions assessed the participants’ knowledge of the responsibilities of AI in radiology (Table 4). It was apparent from this table that 88% of the participants agreed with having the results verified before they were approved. Interestingly, radiographers in the X-ray department were more likely to answer differently compared with the CT/MRI staff and radiologists, (Welch’s F (4, 342) = 4.54, *p* = 0.001). Of the study population, 65% agreed that patients should be aware of using AI, and their consent should be obtained. Younger participants were more likely to agree with this compared with their older counterparts (OR: 1.5, CI: 1.1 to 2, *p* < 0.05). When patients were asked about whether AI contributes to the quality of patient care, only 61% agreed. Further analysis of this question revealed that the males and the older group were more likely to agree with this statement compared with the females and the younger group, (OR: 1.5, CI: 1.1 to 2, *p* < 0.05 and OR: 0.6, CI: 0.46 to 0.85, *p* < 0.05, respectively). Similarly, 62% of the participants thought AI could make radiologists and physicians more efficient in their practice and the males and the older group were more likely to agree, (OR: 1.5, CI: 1.1 to 2, *p* < 0.05 and OR: 0.6, CI: 0.4 to 0.77, *p* < 0.001, respectively).

### 3.6. AI Validation

The final set of questions assessed the participants’ knowledge on the validation of AI in radiology (Table 5). Of the 714 participants, 65% of them felt that the use of AI made medical services more accurate. Further analysis showed that the older group were more likely to agree with this statement, (OR: 0.7, CI: 0.52 to 1, *p* < 0.05). Moreover, the majority of the participants (74%) did not think the use of AI made the medical services more humane. Male respondents compared to their counterparts were almost twice more likely to agree with this, (OR: 1.7, CI: 1.2 to 2.4, *p* < 0.05). In addition, 82% of the participants felt that AI must be included in the curriculum of medical and allied health colleges. In addition, 86% of the participants agreed that AI would be essential in the future. Further analysis showed that the responses of the participants from the eastern region were significantly different from those of their counterparts from the southern region, (Welch’s F (4, 191.8) = 4.7, *p* = 0.001). Even though the interaction between machine and men would be one of the most important skills in the future, 89% of the participants thought that it would never replace radiologists.

## 4. Discussion

AI can significantly improve the performance of healthcare providers. This is vital in radiology since radiology is one of the largest generators of big data. The transformation to AI can assist in reducing the workload of healthcare providers as well as improving image acquisition and evaluation. However, there is a lack of studies on how radiology specialists and radiologists would perceive such a transformation, particularly in Saudi Arabia. To our knowledge, this is the first publication of this kind to include all radiology staff from all over Saudi Arabia.

The findings presented in this study showed that the majority of the participants have read or heard about AI, mainly through the media (63%). The study confirmed that the media played an important role in shaping public perception. These results reflected those of Goldberg et al., who also found that the media characterized public perspectives regarding the use of AI in radiology [14]. Approximately 60% of our sample claimed that treatment was made more effective by using AI. This is consistent with the analysis by Gong et al., in which more than 90% of Canadian medical students accepted that AI would improve the potential of radiologists and make them more effective [15].

In our research, 80% of the respondents agreed that AI technologies must be validated in a well-established clinical setting. The medical societies had to address concerns about how to test AI software for therapeutic efficacy and safety prior to the implementation of a large AI clinical application [16]. Various investigations examined and contrasted AI resources to extremely advanced activities, such as X-ray pneumonia diagnosis or mammogram breast cancer screening [17]. However, these algorithms were constrained in some anatomical regions to particular diseases. Nowadays, the superiority of computers over humans is not a matter of debate; rather, the question is centered around how the practice of medicine can benefit from these capabilities [18]. The detection of pulmonary nodules and wrist fractures has been shown to be enhanced by the use of AI [19,20,21]. Nevertheless, this may lead to a concern in regard to biasing the physician’s decision [22].

In addition, we identified differences between the attitudes of the radiology staff concerning the securing the patient’s approval prior to the use of AI. They all agreed with the statement that patients should be informed. Because discrepancies between the evaluation of radiologists and AI could lead to patients’ irritation and doubt, this question is still one of the key areas concerning the use of AI in radiology [16]. Moreover, 82% of the respondents stated that AI should be integrated into the medical curricula. This is consistent with a study by Dos Santos et al., where 77% believed that AI will revolutionize radiology and should be included in medical education [23,24].

Gong et al. showed that about 30% of medical students think that AI will replace radiologists in the future [15]. This differed against the findings presented in this study, where only 11% collectively shared this opinion. There were similarities between the attitudes expressed by the participants in this study and those described by Dos Santos et al., where 83% of the respondents disagreed with the statement that AI would replace human radiologists [23]. Another recently published study found that radiologists have a positive attitude toward the implementation of AI and are not concerned that AI will replace them [25]. At the same time, AI may replace human medical expertise in specific and repetitive tasks, such as detecting disease indicators in images [26]. Finally, the majority of the participants (84%) agreed with the fact that the interaction between man and machine is one of the most important skills in the future. This is consistent with the findings of Davenport and Kalakota, who showed that only those who refused to work alongside AI will lose their jobs [27].

## 5. Conclusions

This study is the first Saudi survey to assess the awareness and attitude of the radiology community in regard to AI applications. Because AI plays a role in image recognition and acquisition and also improves the support tools for radiology decisions, it is important to ensure that radiologists and radiographers have at least a minimum understanding of the technology. Our finding demonstrated the participants possessed an acceptable level of knowledge regarding AI technology. It also showed that the interaction between man and machine would be one of the most important skills in the future, and thus, AI applications should be included in the curriculum of medical students. A natural progression of this work is to analyze the impact of AI on routine radiology procedures and the challenges facing its clinical implementation in Saudi Arabia.

## Figures and Tables

**Figure 1 healthcare-09-00834-f001:**
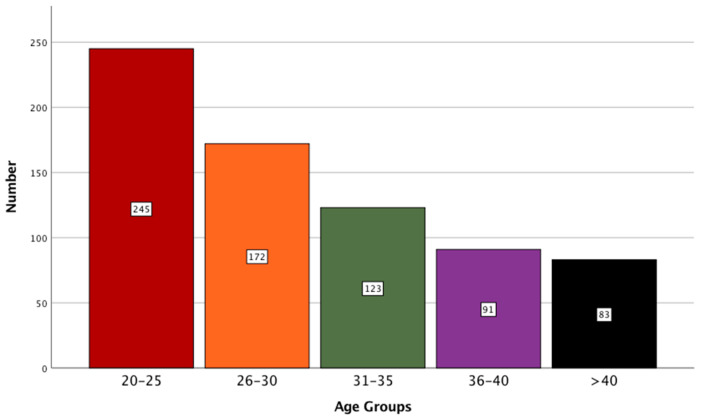
A bar graph showing the distribution of participants based on their age.

**Figure 2 healthcare-09-00834-f002:**
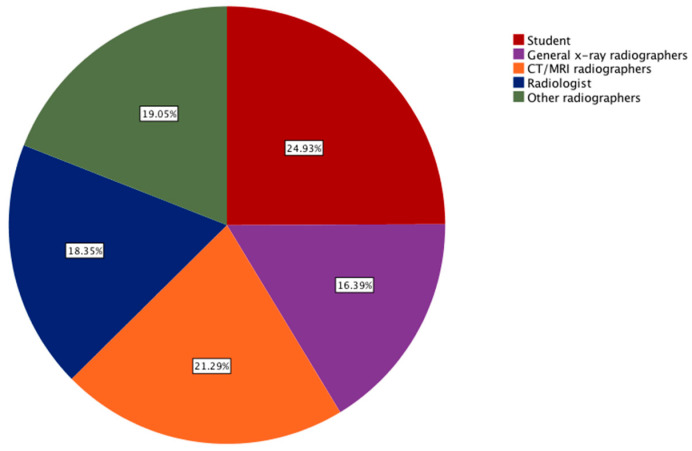
A pie graph showing the distribution of participants based on their professions.

**Figure 3 healthcare-09-00834-f003:**
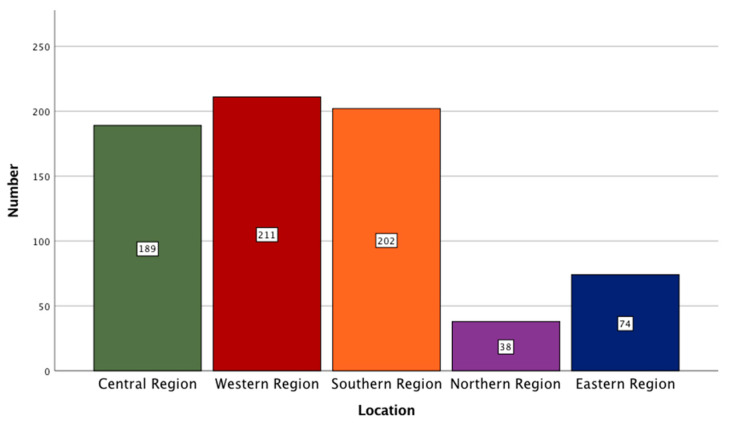
A bar graph showing the distribution of participants based on their locations.

**Table 1 healthcare-09-00834-t001:** Awareness responses.

Questions	Answers	No.	%
Have you ever read or heard about artificial intelligence and its role in radiology?	Agree	437	61.2
	Disagree	277	38.8
Is your knowledge about artificial intelligence based on what is published in the media?	Agree	453	63.4
	Disagree	261	36.6
Are you keen to attend conferences and courses about artificial intelligence in radiology?	Agree	311	43.6
	Disagree	403	56.4
Are you involved in research projects on developing applications of artificial intelligence?	Agree	170	23.8
	Disagree	544	76.2
Does artificial intelligence contribute in the preparation of radiographic reports?	Agree	389	54.5
	Disagree	325	45.5

**Table 2 healthcare-09-00834-t002:** Awareness responses.

Questions	Answers	No.	%
Artificial intelligence contributes to obtain high-quality images	Agree	509	71.3
	Disagree	205	28.7
Artificial intelligence contributes to the archiving system (PACS)	Agree	576	80.7
	Disagree	138	19.3
Artificial intelligence contributes toward the selection of appropriate scanning protocols for CT/MRI imaging	Agree	456	63.9
	Disagree	258	36.1
Is the weakness in training new graduates on artificial intelligence skills one of the greatest obstacles to the application of artificial intelligence in the work environment?	Agree	493	69
	Disagree	221	31
Will the application of artificial intelligence threaten some radiological professions ?	Agree	312	43.7
	Disagree	402	56.3

**Table 3 healthcare-09-00834-t003:** Outcomes responses.

Questions	Answers	No.	%
Can the result of radiographic examination by the artificial intelligence be considered reliable in routine cases?	Agree	285	39.9
	Disagree	429	60.1
The results of radiographic examination by the artificial intelligence need to be verified by the radiologist	Agree	568	79.6
	Disagree	146	20.4
Does conflict in results and interpretation between the various artificial intelligence algorithms and the opinion of the doctor cause stress and anxiety for the patient	Agree	425	59.5
	Disagree	289	40.5
The radiologists are the only ones responsible for the results of the utilization of artificial intelligence	Agree	182	25.5
	Disagree	532	74.5
Shared responsibility must be applied between artificial intelligence companies, hospitals, and international organizations regarding the results of using of artificial intelligence	Agree	595	83.3
	Disagree	119	16.7

**Table 4 healthcare-09-00834-t004:** Responsibilities responses.

Questions	Answers	No.	%
The validity of the results from artificial intelligence must be verified	Agree	628	88
	Disagree	86	12
The patient should be aware of the use of artificial intelligence, and his or her consent should be obtained	Agree	463	64.8
	Disagree	251	35.2
The use of artificial intelligence contributes to the improvement of patient care	Agree	437	61.2
	Disagree	277	38.8
Should information issued about artificial intelligence be available only to radiologists?	Agree	238	33.3
	Disagree	476	66.7
Does the use of artificial intelligence enhance the capabilities of both physicians and radiologists and make them more efficient?	Agree	445	62.3
	Disagree	269	37.7

**Table 5 healthcare-09-00834-t005:** Validation responses.

Questions	Answers	No.	%
The use of artificial intelligence makes medical services more accurate	Agree	463	64.8
	Disagree	251	35.2
The use of artificial intelligence makes medical services more humane	Agree	189	26.5
	Disagree	525	73.5
Artificial intelligence must be included in the curriculum and training of medicine and health sciences colleges	Agree	585	81.9
	Disagree	129	18.1
Artificial intelligence cannot dispense with the role of physician and radiologist but makes a change in the work environment	Agree	635	88.9
	Disagree	79	11.1
The interaction between man and machine will be one of the most important medical skills in the future.	Agree	617	86.4
	Disagree	97	13.6

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
