# Peer review of "Radiology Community Attitude in Saudi Arabia about the Applications of Artificial Intelligence in Radiology"

_healthcare, 2021, doi:10.3390/healthcare9070834_

Round 1

Reviewer 1 Report

This is an interesting paper that deals with a timely topic.

I have few suggestions.

In the abstract and the Results I would delete "This was not surprising, as" as this is more like Discussion.

The authors should use "AI" consistently in the paper after the first introduction of this acronym.

In the M&M section, the authors should indicate how many questionaires were sent to have the % of responders.

I suggest that the authors use color graphs in figs. 1, 2 and 3.

I suggest that the authors use the same P for p values consistently

OR, 95%CI and P values should be into parentheses in the Results section

I suggest that the authors add proportions to % in the Tables

The presentation of the references should follow a more classical format.

Some references must be updated

I suggest the following one for Lung Nodules

 Blanc D, Racine V, Khalil A, Deloche M, Broyelle JA, Hammouamri I, et al. Artificial intelligence solution to classify pulmonary nodules on CT. Diagn Interv Imaging 2020;101(12):803-810.

Additional references about similar approaches should be added for further comparison with European surveys

Coppola F, Faggioni L, Regge D, Giovagnoni A, Golfieri R, Bibbolino C, Miele V, Neri E, Grassi R. Artificial intelligence: radiologists' expectations and opinions gleaned from a nationwide online survey. Radiol Med 2021;126(1):63-71.

Waymel Q, Badr S, Demondion X, Cotten A, Jacques T. Impact of the rise of artificial intelligence in radiology: what do radiologists think? Diagn Interv Imaging 2019;100(6):327-336.    Huisman M, Ranschaert E, Parker W, Mastrodicasa D, Koci M, Pinto de Santos D, et al.  An international survey on AI in radiology in 1,041 radiologists and radiology residents part 1: fear of replacement, knowledge, and attitude. Eur Radiol 2021;doi: 10.1007/s00330-021-07781-5.   I suggest to delete references with non peer reviewed articles such as ArXiv

Reviewer 2 Report

The current manuscript presents the attitude within the radiology community in Saudi Arabia towards artificial intelligence. The paper is based on a survey from 714 answers. The paper is well written and very timely. However, the main concern for this paper is lack of detailed information on how the answers are related to the specific professions (e.g. radiologist, technician etc), which significantly reduces the value of the report.

Comments

Introduction

p2.

The introduction is very long. There are many possibilities to shorten the Introduction without loosing important material. Especially, the paragraph starting with “The most beneficial AI tools …” can be significant shorter, but the whole page 2 would benefit from a shorter more stringent language.

Material and Methods

p3.

It is not well described how the survey was distributed. How many surveys were sent out? To whom? How were the respondents selected for receiving a survey? The text only refers to “target group” but no numbers or details are given.

Results

p3

The number of non-responders is not given.

The qualifications are given as bachelor, masters and PhD. But relevant data are their professions. To be able to use the data in this report one need to know the respondents as radiographers, technicians, engineers, radiologists etc. It is fine to report for students, but then one need to specify if it is radiology (medical student) or a student aiming to be a technician.

Lack of clarity of profession is an issue for many of the results. See for example fig 2. Are this pie chart for radiologists only? For the category “specialized in MR/CT”, what profession is this category referring to?

This issue also affects several questions. For example (Table 1) “Does artificial intelligence contribute in the preparation of radiographic reports?”

It does not make sense to report the answers to this question without specifying the profession. The same is valid for all questions specifically asking about “radiologists”, which are many in the survey.

Round 2

Reviewer 2 Report

The authors have as stated “incorporated changes to reflect most of suggestions provided by the reviewers”. That is acknowledged.

In the Materials and Methods section it is added “and the link was distributed across Saudi Arabia through emails, WhatsApp groups and Twitter”.

How were the email addresses selected? Using Twitter I assume anybody in the world can answer the survey? To my opinion a survey distributed using such vague methods does not rectify a publication in a scientific journal.

Author Response

First of all, I appreciate the reviewer’s concern regarding the use of email and Twitter as a means for collecting the data. 

In addition, the research objective is to look at the knowledge and awareness of radiology staff in regard to the use of AI in radiology in Saudi Arabia. To accomplish this, we have sent emails to only our emailing list (already known radiologists and radiographers), and they were asked to distribute the questionnaire across their colleagues who work in their departments. 

In order to reach out to more participants from different cities, Twitter was introduced and only the targeted group were encouraged to take a role in the questionnaire. 

Using Twitter in research is not limited to our study. Many researchers have used the social media in their research. See for example: 

Finally, the number of participants is not huge that would raise the concern of having other than the targeted group participating in the research. 

I hope the above points satisfy the reviewer’s concerns.